# Transcriptomic and Metabolomic Analysis Reveal the Effects of Light Quality on the Growth and Lipid Biosynthesis in *Chlorella pyrenoidosa*

**DOI:** 10.3390/biom14091144

**Published:** 2024-09-10

**Authors:** Tingting Zhu, Ge Guan, Lele Huang, Lina Wen, Linxuan Li, Maozhi Ren

**Affiliations:** 1Institute of Urban Agriculture, Chinese Academy of Agricultural Sciences, Chengdu National Agricultural Science and Technology Center, Chengdu 610213, China; zhutingting@caas.cn (T.Z.); huanglele63@outlook.com (L.H.); 2School of Agricultural Science, Zhengzhou University, Zhengzhou 450001, China; geguan@gs.zzu.edu.cn (G.G.); linawen@gs.zzu.edu.cn (L.W.)

**Keywords:** light quality, biomass, lipid biosynthesis, omics analysis, *Chlorella pyrenoidosa*

## Abstract

Light quality has significant effects on the growth and metabolite accumulation of algal cells. However, the related mechanism has not been fully elucidated. This study reveals that both red and blue light can promote the growth and biomass accumulation of *Chlorella pyrenoidosa*, with the enhancing effect of blue light being more pronounced. Cultivation under blue light reduced the content of total carbohydrate in *Chlorella pyrenoidosa*, while increasing the content of protein and lipid. Conversely, red light decreased the content of protein and increased the content of carbohydrate and lipid. Blue light induces a shift in carbon flux from carbohydrate to protein, while red light transfers carbon flux from protein to lipid. Transcriptomic and metabolomic analysis indicated that both red and blue light positively regulate lipid synthesis in *Chlorella pyrenoidosa*, but they exhibited distinct impacts on the fatty acid compositions. These findings suggest that manipulating light qualities can modulate carbon metabolic pathways, potentially converting protein into lipid in *Chlorella pyrenoidosa*.

## 1. Introduction

*Chlorella pyrenoidosa* is a unicellular freshwater alga belonging to the green algae class, with a diameter ranging from 3 to 8 μm. *Chlorella pyrenoidosa* exhibits rapid growth and widespread distribution, serving as an efficient photosynthetic organism rich in proteins, lipids, various vitamins, essential amino acids, and trace elements [1,2], which can meet the nutritional needs of both humans and animals. Chlorella has been listed as a green nutritional source for the 21st century by the FAO. The protein content of *Chlorella pyrenoidosa* exceeds 50% of its cell dry weight [3,4], making it a significant plant-based protein source. It can be utilized as a raw material for human food and as protein feed for animals. *Chlorella pyrenoidosa* also exhibits the functions of regulating blood lipids, anti-aging, enhancing immune function and regulating intestinal microflora [5,6,7], which is an excellent nutritional supplement.

Light not only provides energy for the photosynthetic or mixotrophic growth of microalgae, but also serves as a signaling molecule regulating the biosynthesis of crucial metabolites [8,9]. Light plays a crucial role in achieving the efficient cultivation of microalgae and the abundant accumulation of target products. Light intensity and quality are among the most environmental factors affecting the growth and biochemical compositions of microalgae, such as photosynthesis, lipid production, and metabolite accumulation [10,11]. Light-emitting diodes (LEDs), which have a higher energy efficiency and intensity compared to conventional fluorescent light, are widely used in microalgae cultivation [12,13]. Algal cell growth and metabolic pathways can be regulated by using LED light with specific wavelengths [13,14,15,16,17,18]. Blue light can promote chlorophyll synthesis and increase the biomass and lipid content of *Chlorella vulgaris.* The highest lipid content can reach 23.5% under blue light, whereas it is limited to 20.9% under white light [19]. Previous studies also suggest that blue light stimulates the synthesis of carotenoids, including lutein and astaxanthin [16,20]. Some studies have also found that red light is the best light source for microalgae biomass accumulation [15,21,22]. Additionally, red light promotes lipid, phycobiliprotein, carotenoid, and β-carotene synthesis [23,24]. The growth and metabolic processes of different microalgae under various light qualities are not entirely consistent. Therefore, it is of great significance to explore stable, efficient, and cheap light quality culture conditions for efficient and high-quality production in specific microalgae species.

Some studies show that dynamically adjusting the wavelength distribution can further enhance productivity in microalgae. Red light promotes cell growth and blue light induces astaxanthin accumulation in *Haematococcus pluvialis*. Consequently, a high biomass and high astaxanthin concentration can be obtained by culturing with red light first and then blue light [20]. Similarly, blue light stimulates cellular enlargement, whereas red light promotes cell division in *Chlorella vulgaris*. The strategy of first culturing with blue light and then switching to red light significantly increases the biomass and lipid production of *Chlorella vulgaris* [25]. These findings suggest that dynamic light conditions during cultivation can achieve metabolic modulation in microalgae, and provide a new way to improve biomass and target products of microalgae. However, the effects of different light qualities on the growth and metabolism of *Chlorella pyrenoidosa* still need further study.

The protein content of *Chlorella pyrenoidosa* is high, while its lipid content is relatively low. The possibility of manipulating the carbon metabolic pathways using various light qualities to convert the proteins of *Chlorella pyrenoidosa* into lipids, thereby facilitating carbon flux transition from protein source to triacylglycerol (TAG) sink, is not yet fully understood. This study reveals that different light qualities distinctly affect the growth and biomass accumulation of *Chlorella pyrenoidosa*. In comparison to white light, both red and blue light significantly enhance growth and biomass accumulation, with blue light showing a more pronounced effect in *Chlorella pyrenoidosa*. Transcriptomic analysis indicates that red and blue lights enhance the expression of most photosynthesis-related genes, consistent with the phenotype observed, where blue light induces a greater number of genes and higher expression levels. Furthermore, blue light decreases the total sugar content while increasing protein and lipid content in *Chlorella pyrenoidosa*. Conversely, red light decreases the protein content but increases the total sugar and lipid content. Transcriptomic and metabolomic correlation analysis reveal that both blue and red lights can enhance the expression levels of fatty acid synthesis genes and the fatty acid content. Promoter analysis identifies an abundance of light-responsive elements in the promoter regions of fatty acid synthesis genes, implying that transcription factors in the light signal pathway may directly target and activate these genes. These findings indicate that different light qualities can regulate the carbon metabolic pathways in *Chlorella pyrenoidosa*. This provides new insights into the optimization of intracellular carbon distribution, and offers theoretical underpinnings for efficient and high-quality production in *Chlorella pyrenoidosa*.

## 2. Materials and Methods

### 2.1. Algal Strain and Culture Conditions

The XL12 strain of *Chlorella pyrenoidosa* was collected and deposited in our laboratory. The strain was cultured in blue-green (BG-11) liquid medium containing 20 g·L^−1^ glucose under white light with a continuous light intensity of 200 μmol.m^−2^·s^−1^ at 25 ℃ and 180 rpm.

The light quality experiments were conducted in 500 mL glass conical bottles with the same inoculation density (OD_680_ = 0.3), and were allowed to grow under the same conditions as mentioned above. Light was provided by a custom-built illumination system, which allows us to monitor and regulate the light intensity and quality. Red LED light (λmax = 660 nm) and blue LED light (λmax = 460 nm) were used to provide monochromatic red and blue lighting, respectively. Meanwhile, white LED light (red: green: blue = 3:6:1) was used as control. The light intensity of all experiments was set at 200 μmol.m^−2^·s^−1^. The light intensity of the photosynthetically available radiation (PAR) region was measured in each flask by using a spectral illuminometer (PLA-30, Everfine, Hangzhou, China). The cell density at an optical density of 680 nm (OD_680_) was measured with a spectrophotometer (Biotek EpochTM2, BioTek, Winooski, VT, USA).

### 2.2. Measurement of Microalgae Biochemical Compositions

The chlorophyll was extracted and measured as described in a previous study [26]. Briefly, the chlorophyll content was measured using a spectrophotometric method. Chlorophyll a and chlorophyll b were measured at 645 nm and 663 nm, respectively.

Total carbohydrate content, total protein content, and total lipid content were analyzed as described in a previous study [27].

### 2.3. Free Fatty Acid Content Analysis

The XL12 strain was cultured in BG-11 liquid medium with white light until OD_680_ = 2.0, and was then continuously illuminated with blue, red, and white light for 24 h. The precipitate was collected by 9000× *g* centrifugation at 4 °C. The experiment was repeated three times. Fatty acid compositions and contents were detected by MetWare biotechnology Co., Ltd. (Wuhan, China) based on the Agilent 7890B-7000D GC-MS/MS platform.

### 2.4. Transcriptome Sequencing, Assembly, and Functional Annotation

The total RNA of the XL12 strain treated with red light, blue light, and white light for 24 h was extracted using an RNA extraction kit (TIANGEN, Beijing, China). The integrity and concentration of RNA samples were detected using an Agilent 5400 analyzer (Agilent, Santa Clara, CA, USA), while DNA contamination was examined with agarose gel electrophoresis. The Collibri Stranded RNA Library Prep Kit (Invitrogen, Carlsbad, CA, USA) was used to construct the library, and the library was sequenced on an Illumina Novaseq platform by Novogene Co., Ltd. (Beijing, China). Transcriptome assembly was accomplished based on the clean reads using Trinity software (version 2.6.6) [28]. The functional annotation of transcripts was performed using the Nr, Nt, Pfam, KOG/COG, Swiss-prot, KEGG, and GO databases.

Differentially expressed genes (DEGs) were analyzed by using the DESeq2 R package (version 1.20.0) [29]. Padj < 0.05 and | Log_2_ (Fold Change) | > 1 were set as the threshold values of gene differential expression. Gene Ontology (GO) functional enrichment and Kyoto Encyclopedia of Genes and Genomes (KEGG) pathway enrichment analysis of DEGs were performed by using GOseq (version 1.3.2) and KOBAS software (version 3.0).

### 2.5. Validation of Transcriptome Data Using Quantitative Real-Time PCR (qRT-PCR)

RNA that was processed in the same batch as transcriptome sequencing was selected for qRT-PCR. CpActin was used as a reference gene (Appendix A). qRT-PCR was conducted with a SYBR Green SuperReal PreMix Plus Kit (TIANGEN, Beijing, China) and a QuantStudio 3 Real-Time PCR System (ThermoFisher, Waltham, MA, USA). Reactions were performed in a final volume of 20 µL containing 10 µL of 2 × SuperReal PreMix Plus, 50 ng cDNA, and 0.3 µM each for forward and reverse primers. The following program was used: 95 °C for 15 min; followed by 40 cycles of 95 °C for 10 s and 60 °C for 32 s each cycle; and a melt curve stage of 95 °C for 15 s, 60 °C for 1 min, and 95 °C for 5 s. Relative transcript levels were calculated according to the following formula: 2^−ΔΔCt^. Three biological replications were performed.

### 2.6. Untargeted Metabolome Analysis

The XL12 strain was treated with red light, blue light, and white light for 24 h. The precipitate was collected by 9000× *g* centrifugation at 4 °C, and resuspended with prechilled methanol [30]. The supernatant was injected into the LC-MS/MS system for analysis. UHPLC-MS/MS analyses were performed using a Vanquish UHPLC system (ThermoFisher, Waltham, MA, USA) coupled with an Orbitrap Q Exactive TM HF mass spectrometer (ThermoFisher, Waltham, MA, USA) in Novogene Co., Ltd. (Beijing, China). Quality control (QC) samples are composed of equal volumes of the test samples. Six QC samples were run before test samples were run.

The raw data files generated by UHPLC-MS/MS were processed using Compound Discoverer 3.1 software (version 3.1) to perform peak alignment, peak picking, and quantitation for each metabolite. The normalized data were used to predict the molecular formula based on additive ions, molecular ion peaks, and fragment ions. Then, peaks were matched with the mzCloud, mzVault, and MassList databases to obtain the accurate qualitative and relative quantitative results.

These metabolites were annotated using the KEGG database (http://www.genome.jp/kegg/pathway accessed on 19 April 2023), HMDB database (https://hmdb.ca/metabolites accessed on 19 April 2023), and LIPIDMaps database (http://www.lipidmaps.org/ accessed on 19 April 2023). The metabolites with VIP > 1, *p*-value< 0.05, and | Log_2_ (Fold Change) | > 1 were considered to be differential accumulated metabolites (DAMs).

### 2.7. Lipid Bodies (LDs) Staining

Nile Red (Coolaber, Beijing, China) and BODIPY 505/515 (Sigma Aldrich, St. Louis, MO, USA) were used for the staining of the intracellular lipid bodies of *Chlorella pyrenoidosa*, as described in a previous study [31]. The XL12 strain was treated with red light, blue light, and white light for 24 h, and LDs in algal cells were stained with 0.1 mg. mL^−1^ Nile Red or 1 μM BODIPY 505/515. The samples were imaged using a fluorescence microscope (Axio Imager M2, ZEISS, Oberkochen, Germany).

### 2.8. Statistical Analysis

Two-way ANOVA was performed using the IBM SPSS statistics software (version 20) to investigate the statistical differences between control and other samples. Different letters represent significant differences between treatments (Tukey, *p* < 0.05) in the two-way ANOVA. All experiments had at least three biological replicates, and experimental results were described as mean ± standard deviation.

## 3. Results and Discussion

### 3.1. The Effect of Different Light Qualities on the Growth of Chlorella pyrenoidosa

Previous studies have shown that microalgae chlorophyll mainly absorbs blue and red light [32]. To verify the effects of different light qualities on the growth and biochemical composition of *C. pyrenoidosa*, the OD_680 nm_, protein content, and chlorophyll content of *C. pyrenoidosa* were measured under red, blue, and white light conditions (Figure 1). With the increase in cultivation time, *C. pyrenoidosa* showed a faster growth rate and greener color under blue light, and also exhibited the most rapid increases in cell density, chlorophyll, and protein content. The effect of red light was weaker than that of blue light but was better than that of white light (Figure 1). This result is consistent with previous reports [33]. *C. pyrenoidosa* showed a faster growth rate under blue light, which was about twice as fast as that under white light (Figure 1B). Chlorophyll and protein contents under blue light were also significantly higher than those under white light (Figure 1C,D). After 6 days of cultivation, protein concentrations reached 1.91 mg·mL^−1^ under blue light, which was 1.71-fold higher than that under white light. These findings indicate that blue light was more suitable for the cultivation of *C. pyrenoidosa*.

Although blue and red lights were more favorable for the growth and biomass accumulation of *C. pyrenoidosa* (Figure 2A), there are significant differences in the accumulation of metabolites. Cultivation under blue light reduced the content of total carbohydrate in *C. pyrenoidosa*, while increasing the contents of protein and lipid. Conversely, red light decreased the content of protein and increased the contents of carbohydrate and lipid. Notably, both blue and red lights can increase the lipid content in *C. pyrenoidosa* (Figure 2B–D). The highest carbohydrate and lipid contents reached 23.15% and 21.14%, respectively, following cultivation for 2 days under red light conditions, which were 1.74- and 1.45-fold higher than that of white light. Additionally, the highest protein content reached 57.51% after cultivation for 2 days under blue light conditions, which was 12.88% higher than that of white light and 19.51% higher than that of red light. Blue light induced a shift in carbon flux from carbohydrate to protein, while red light transferred carbon flux from protein to lipid. These results indicated that different light qualities can regulate carbon metabolism pathways and achieve different carbon metabolism flows in *C. pyrenoidosa*. During the cultivation of *C. pyrenoidosa*, continuously cultivating with red light for two days before harvest can significantly increase carbohydrate and lipid contents, and cultivating with blue light for two days before harvest significantly increases protein content. Thus, red light can be used as an accelerator for high carbohydrate and lipid contents, and blue light is suitable for the high biomass and protein accumulation of *C. pyrenoidosa* in future studies.

### 3.2. Transcriptomic Analysis of C. pyrenoidosa under Different Light Qualities

To investigate the molecular mechanisms of different light qualities affecting the growth and metabolism of *C. pyrenoidosa*, transcriptome sequencing was performed on *C. pyrenoidosa* under red, blue, and white light conditions for 24 h. A total of 1570 differentially expressed genes (DEGs) were detected between the blue light group and the white light group, of which 857 DEGs were upregulated and 713 DEGs were downregulated. In total, 1572 DEGs were found between the red light group and the white light group, with 941 DEGs upregulated and 631 DEGs downregulated (Figure 3A). The Venn diagram showed 1099 DEGs overlapped between the red vs. white and blue vs. white groups (Figure 3B). Cluster analysis indicated that red and blue light treatments resulted in changes in the expression levels of many genes compared to white light (Figure 3C).

Our study indicated that red and blue light mainly promote the growth of *C. pyrenoidosa* and the accumulation of metabolites. Thus, this study primarily focused on analyzing the biological functions of upregulated genes under red and blue light conditions. The GO enrichment analysis results indicated that the upregulated DEGs were significantly enriched in the following GO terms: thylakoid, photosynthesis, oxidoreductase activity, ribosome, and lipid metabolic process in the blue vs. white light group (Appendix A). These GO terms are closely related to photosynthesis, protein, and lipid synthesis. In the red vs. white light group, the upregulated DEGs were significantly enriched in the following GO terms: oxidoreductase activity, lipid binding, photosynthesis, catalytic activity, and lipid metabolic process, which are related to photosynthesis and lipid synthesis. Moreover, the autophagy term was also significantly enriched among the upregulated GO terms in the red vs. white light group. Some studies have demonstrated that autophagy is a key pathway for protein degradation [34], which may be one of the reasons for the reduced protein content in *C. pyrenoidosa* under red light. Protein degradation resulted in a shift of carbon metabolic flux to carbohydrates and lipids under red light, thereby increasing the carbohydrate and lipid content of *C. pyrenoidosa*. The GO enrichment analysis results suggested that different light qualities affect various biological processes in *C. pyrenoidosa*.

KEGG enrichment analysis was performed for upregulated DEGs in *C. pyrenoidosa*. The top three most significantly enriched KEGG pathways in the blue vs. white light group were photosynthesis–antenna proteins, photosynthesis, and fatty acid biosynthesis (Figure 3D). Photosynthesis–antenna proteins, fatty acid biosynthesis, and sulfur metabolism were most significantly enriched in KEGG pathways in the red vs. white light group (Figure 3E). A total of 44 genes were differentially expressed in the photosynthetic process, of which 43 genes were upregulated and 1 gene was downregulated in the blue vs. white light group (Appendix A). There were 29 DEGs involved in photosynthesis in the red vs. white light group, among which 28 genes were upregulated and 1 gene was downregulated. Importantly, the number of DEGs and the expression levels of DEGs under blue light were higher than those under red light, indicating that blue light is more conducive to the photosynthesis of *C. pyrenoidosa*. This is consistent with the growth phenotype of *C. pyrenoidosa*. In line with the GO terms, ribosomes were also significantly enriched in the upregulated KEGG pathways in the blue vs. white light group (Figure 3D), with 37 upregulated genes including ribosomal large subunit, small subunit, and translation initiation and elongation factors (Appendix A), indicating that blue light enhanced the synthesis of protein in *C. pyrenoidosa*. This should be the reason why the biomass and protein content of *C. pyrenoidosa* under blue lighting were obviously higher than those under red and white light conditions.

### 3.3. Metabolomic Analysis of C. pyrenoidosa under Different Light Qualities

The algae cells that were prepared in the same batch as transcriptome sequencing were selected for untargeted metabolome analysis. Positive and negative ions were combined to enhance the accuracy and reliability. Principal components analysis of the metabolites indicated that the biological replicates within the three treatment groups were relatively clustered together. PC1 accounted for 24.61% of the total variance, while PC2 accounted for 20.29% (Figure 4A). In addition, QC samples are clearly distinguished from different light samples, with all QC samples clustering together. A total of 722 metabolites were detected in all samples, including lipids and lipid-like molecules, organic acids and their derivatives, and organic oxygen compounds. The most abundant category was lipids and lipid-like molecules, with 302 metabolites constituting 41.83% of the total (Figure 4B). Based on the classification results of the annotated metabolites from the lipid maps database, the most abundant category was fatty acids, with 55 detected metabolites, followed by 25 metabolites classified as sterol lipids. Hierarchical clustering analysis results indicated that the three biological replicates in each group cluster together due to a similar metabolite abundance (Figure 4C), which is consistent with the PCA results. Different metabolites were observed in red and blue light conditions compared to white light, particularly in the lipid and lipid-like molecules category.

Based on VIP > 1, *p*-value < 0.05, and | Log_2_ (Fold Change) | > 1, differential accumulated metabolite (DAM) screening was conducted. In total, 140 differential metabolites were found in the blue vs. white light group, of which 67 metabolites were upregulated and 73 metabolites were downregulated (Figure 4D). The red vs. white light group had 82 differential metabolites, with 23 metabolites upregulated and 59 metabolites downregulated (Figure 4E). The category with the most DAMs was lipids and lipid-like molecules. A total of 48 and 29 DAMs related to lipids and lipid-like molecules were detected in the blue vs. white light group and the red vs. white light group, respectively, followed by organic acids and their derivatives, with 23 and 10 DAMs identified, respectively (Figure 4F). These results showed that light quality plays an important role in the lipid metabolism of *C. pyrenoidosa*.

### 3.4. Changes in Lipid Biosynthesis of C. pyrenoidosa under Red And Blue Light Conditions

Our findings suggest that both red and blue light enhance the lipid synthesis of *C. pyrenoidosa*, according to the transcriptomic and metabolomic data. To further analyze the effects of light quality on lipid synthesis, a correlation analysis was performed between transcriptomic and metabolomic data. The significantly enriched KEGG pathways were fatty acid biosynthesis and porphyrin and chlorophyll metabolism between the blue vs. white light group and the red vs. white light group (Appendix A), of which the fatty acid biosynthesis pathway was the most significantly enriched. To determine the effects of red and blue light treatments on lipid metabolics, we further analyzed the DEGs and DAMs associated with fatty acid biosynthesis (Figure 5). A total of 12 DEGs were enriched in the fatty acid biosynthesis KEGG pathway, with all DEGs showing significant upregulation under red and blue light conditions (Appendix A). The upregulated fold of DEGs in the red vs. white light group was higher than that in the blue vs. white light group. This is consistent with the result that the lipid content of *C. pyrenoidosa* cultured in red light is higher than that under blue light.

Promoter analysis revealed that the promoter regions of the 12 DEGs contain multiple light-responsive elements, including G-box, Sp1, and ACE elements (Appendix A), implying that these genes can be induced by light. The bZIP-type transcription factor HY5 is a key positive regulator in the light signaling pathway [35]. HY5 can directly bind to the sequences such as G-box (CACGTT), E-box (CANNTG), and ACE elements (ACGT) on the promoters of light-responsive genes [36,37]. Many G-box and ACE elements can be found in the promoters of fatty acid synthesis genes in *C. pyrenoidosa*, indicating that HY5 may target these genes and promote fatty acid biosynthesis. Previous studies have shown that HY5 targets some genes involved in lipid biosynthesis, such as *DGD1*, *FAD3*, and *FAD6* genes [38]. Similar studies have also indicated that light signal through HY5 was involved in the upregulation of *MGD1* and *DGD1*, which were lipid-related genes [39,40]. These results further confirmed that HY5 regulates lipid metabolism. However, the molecular mechanism by which HY5 regulates lipid synthesis in *C. pyrenoidosa* needs further study.

Transcriptomic and metabolomic correlation analysis results indicated that a total of 10 differentially upregulated metabolites were found in the fatty acid synthesis pathway, including tetradecanoic acid, hexadecanedioic acid, octadecanoic acid, docosahexaenoic acid, linoleic acid, and docosatrienoic acid (Figure 5 and Appendix A). These differentially accumulated metabolites are consistent with the expression pattern of the DEGs in fatty acid biosynthesis. The most upregulated metabolite was docosatrienoic acid in the blue vs. white group, while hexadecanedioic acid was the most accumulated metabolite in the red vs. white group. These results suggested that light quality significantly affects the biosynthesis of fatty acids in *C. pyrenoidosa*. The DEGs and DAMs of the fatty acid biosynthesis pathway showed consistent expression patterns, which further confirmed that red and blue light positively regulated lipid synthesis in *C. pyrenoidosa*.

### 3.5. Effects of Red and Blue Lights on Fatty Acid Compositions of C. pyrenoidosa

To verify the transcriptome and metabolome data, 10 DEGs of fatty acid synthesis were selected for qRT-PCR validation. The qRT-PCR results showed a consistent upregulation trend with the transcriptome data (Appendix A), indicating that the transcriptome data were accurate and reliable. Simultaneously, lipid bodies of *C. pyrenoidosa* with red light, blue light, and white light for 24 h were stained with Nile Red and BODIPY reagents (Figure 6A,B). Lipid body staining results showed that red and blue lights increased lipid content, and red light was more conducive to the lipid accumulation of *C. pyrenoidosa*. Carbohydrate, protein, and lipid content analysis showed that blue light was beneficial to protein synthesis, whereas red light was more beneficial for carbohydrate and lipid synthesis (Figure 6C), which is consistent with the previous results in this study.

The fatty acid compositions of *C. pyrenoidosa* cultured in red and blue lights were analyzed by GC-MS/MS (Figure 6D). The major components of free fatty acids were palmitic acid (C16) and linoleic acid (C18) under red, blue, and white light conditions, which were similar to those of *Chlorella vulgar* [41]. C16 and C18 chains account for over 90% of the total fatty acids, and C18 was the most abundant, constituting more than 55% of the total fatty acids. The main fatty acids of *C. pyrenoidosa* were unsaturated fatty acids, and the main unsaturated fatty acid was C18:2. Red and blue lights promoted the increase in free fatty acids, especially C16- and C18-chain fatty acids. Red light enhanced the synthesis of C16 fatty acids, while blue light was more conducive to the synthesis of C18 fatty acids. The highest content of palmitic acid (C16:0) was 1.37 mg·g^−1^ under red light, which was 1.46 times higher than that under white light. However, the content of linoleic acid (C18:2) was the highest at 1.35 mg·g^−1^ under blue light, with a 31.75% increase compared to white light. In addition, red light was also beneficial for the synthesis of tetradecanoic acid, cis-5,8,11,14,17-eicosapentaenoic acid (EPA), and cis-7,10,13,16,19-docosapentaenoic acid (DHA), whereas blue light was more suitable for the synthesis of octadecanoic acid, cis-9-octadecenoic acid, cis-10-nonadecenoic acid, and cis-11,14-eicosadienoic acid. The results indicated that red and blue light have distinct impacts on fatty acid composition.

## 4. Conclusions

This study reveals that both red and blue light can promote the growth and biomass accumulation of *C. pyrenoidosa*, with the enhancing effect of blue light being more pronounced. Blue light induces a shift in carbon flux from carbohydrate to protein, while red light transfers carbon flux from protein to lipid. Transcriptomic and metabolomic correlation analysis indicate that both red and blue light positively regulate lipid synthesis, but they exhibited distinct impacts on the fatty acid composition. This provides a feasible strategy for enhancing lipid content by carbon flux transition from protein source to lipid sink in *C. pyrenoidosa*.

## Figures and Tables

**Figure 1 biomolecules-14-01144-f001:**
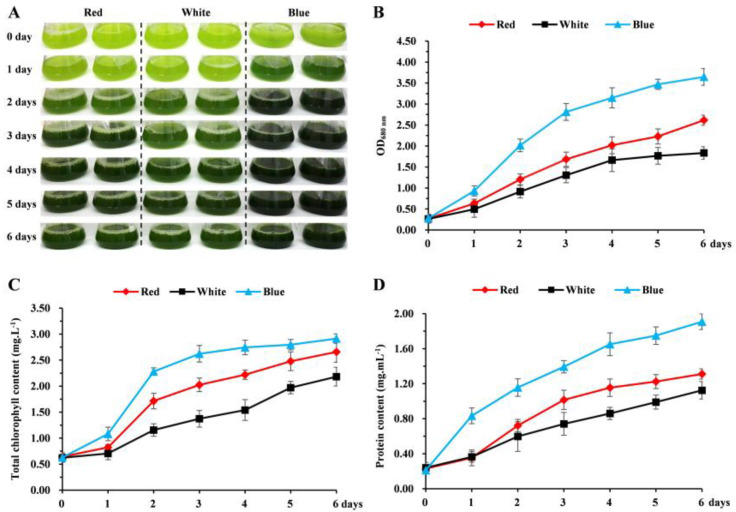
Effects of white, red, and blue LED lights on *C. pyrenoidosa* growth. (**A**) Growth phenotype of *C. pyrenoidosa* under white, red, and blue lighting for 6 days. (**B**) Growth curves of OD_680_ values under white, red, and blue lighting. Chlorophyll (**C**) and protein (**D**) contents of *C. pyrenoidosa* under white, red, and blue lighting.

**Figure 2 biomolecules-14-01144-f002:**
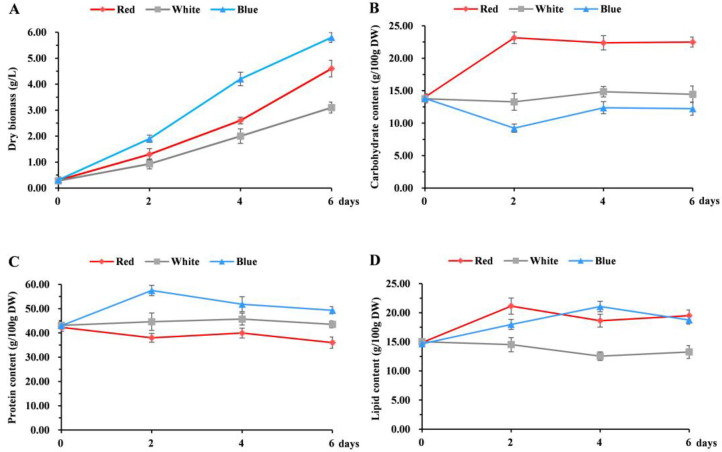
Biomass and metabolite accumulation of *C. pyrenoidosa* under white, red, and blue LED lights. Dry biomass (**A**), carbohydrate content (**B**), protein content (**C**), and lipid content (**D**) of *C. pyrenoidosa* under white, red, and blue lighting for 6 days.

**Figure 3 biomolecules-14-01144-f003:**
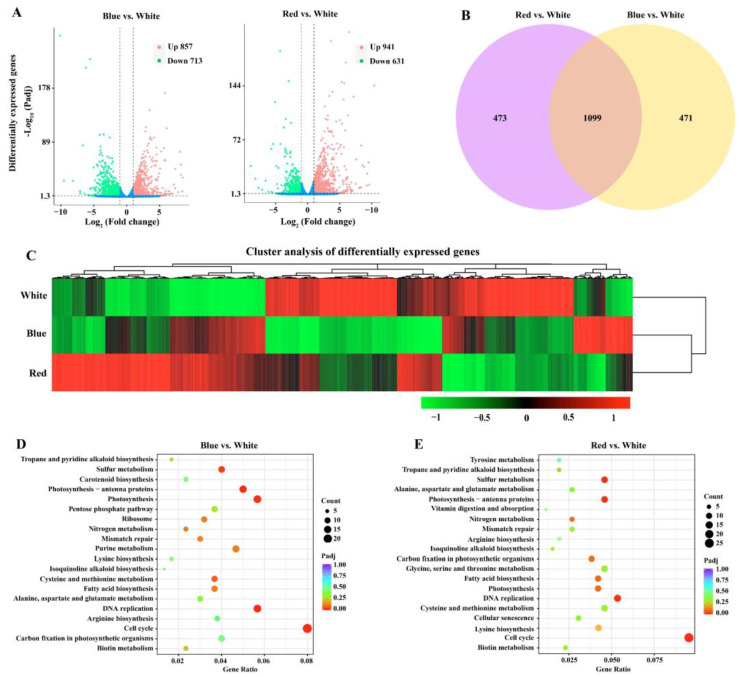
Transcriptome analysis of *C. pyrenoidosa* under red, blue, and white lights. (**A**) The “volcano plot” graph of the DEGs between blue light/red light and the control (white light). (**B**) Venn diagram between red vs. white group and blue vs. white group. (**C**) Cluster analysis of differentially expressed genes for red, blue, and white lights. Red represents high gene abundance, and green represents low gene abundance. The 20 most significantly upregulated KEGG pathways in the blue vs. white group (**D**) and the red vs. white group (**E**).

**Figure 4 biomolecules-14-01144-f004:**
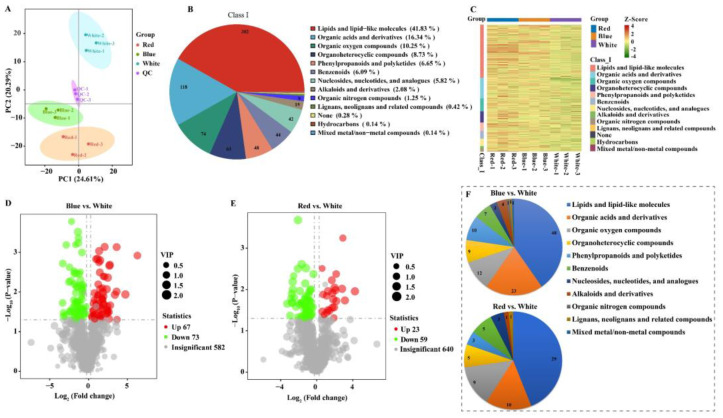
Metabolome analysis of *C. pyrenoidosa* under red, blue, and white lights. (**A**) Principal components analysis (PCA) of red, blue, white, and QC samples. (**B**) Classification of 722 untargeted metabolites. (**C**) Hierarchical cluster analysis of metabolites in red, blue, and white light samples. Each column represents a sample and each row represents a class of metabolites in the sample. A range from green to red indicates a low to high abundance of metabolites. The “volcano plot” graph of the DAMs in the blue vs. white (**D**) and red vs. white (**E**) groups. (**F**) Classification of differential metabolites in the blue vs. white group and red vs. white group.

**Figure 5 biomolecules-14-01144-f005:**
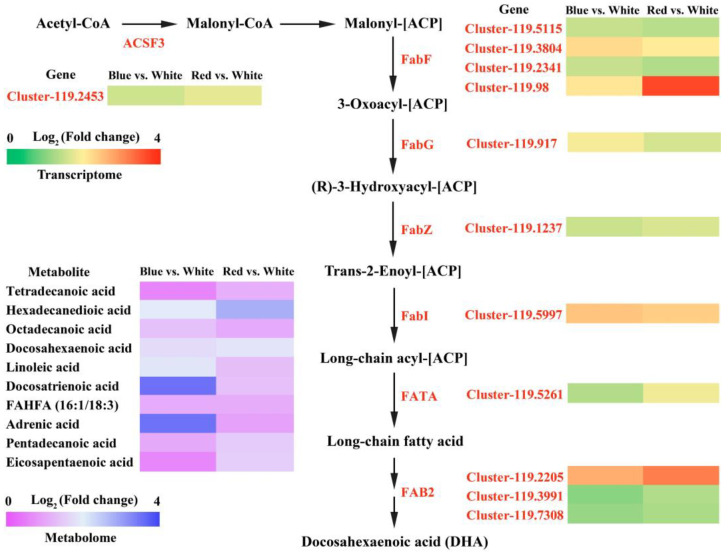
Differentially expressed genes and differential metabolites in the fatty acid biosynthesis pathway. A range from green to red indicates the abundance of differentially expressed genes from low to high, and a range from purple to blue indicates a low to high abundance of differential metabolites.

**Figure 6 biomolecules-14-01144-f006:**
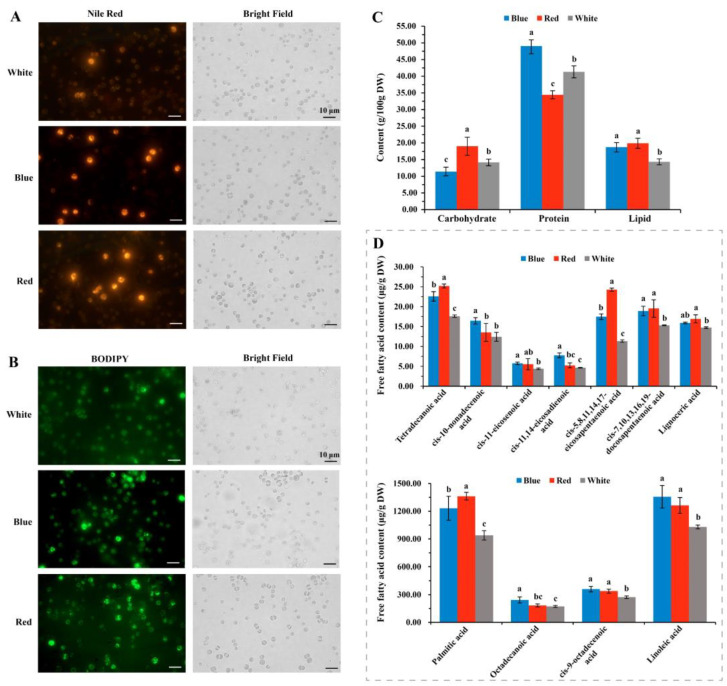
Lipid body staining and fatty acid compositions analysis. The algal cells were stained for lipid bodies with Nile Red (**A**) or BODIPY 505/515 (**B**). (**C**) Carbohydrate, protein, and lipid contents of *C. pyrenoidosa* under continuous white, red, and blue lights for 24 h. (**D**) Analysis of fatty acid compositions in *C. pyrenoidosa* under continuous white, red, and blue lights for 24 h. Different letters represent significant differences between treatments (Tukey, p < 0.05) in the two-way ANOVA.

## Data Availability

Data will be made available on request.

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
