# Peer review of "Transcriptomic and Metabolomic Analysis Reveal the Effects of Light Quality on the Growth and Lipid Biosynthesis in Chlorella pyrenoidosa"

_biomolecules, 2024, doi:10.3390/biom14091144_

Round 1

Reviewer 1 Report

Comments and Suggestions for Authors

Dear authors,

In this study, the authors investigated the effects of red and blue light on the growth, transcriptomic and metabolomic profiles of Chlorella pyrenoidosa. The results revealed that blue and red light incuces a shift in carbon flux from carbohydrate to protein and from protein to lipid, respectively. Because these processes of different micrialgae under various light qualities are not consistent, the results obtained in this study are thought to be important for application of this alga in the future. The manuscripte are well written, however, there are minor concerns to improve for the publication.

L93 etc. Recent paper requires the centrifugation speed as shown in x g not rpm.

L95, 99 etc. The numeric portion of OD680 should be subscripted.

L114 Did you check the contamination of DNA and the integrity of RNA samples. Pleas state clearly.

L128: The experiment conditions of qRT-PCR assay should be written in the text or Supplementary Tables.

Fig.2 and L194. Red light appears to have induced a shift from protein to carbohydrate and lipid. What do you think about the former change?

L218-219: Is it possible that the carbon flux could have been altered by a gene whose expression was reduced?

Fig.3D&E The KEGG enrichment analysis appears to be less consistent with the results in Figure 2. Werer Potein or lipid synthesis related genes enriched under blue or red light? 

Author Response

L93 etc. Recent paper requires the centrifugation speed as shown in x g not rpm.

Response: Thank you for pointing this out. We have converted rpm to g in the revised manuscript.

L95, 99 etc. The numeric portion of OD680 should be subscripted.

Response: Thank you for pointing this out. We have corrected this error in the revised manuscript.

L114 Did you check the contamination of DNA and the integrity of RNA samples. Pleas state clearly.

Response: Thanks for your comments. We used the Agilent 5400 analyzer to detect the integrity and concentration of RNA samples, while DNA contamination was checked using agarose gel electrophoresis. The results showed that RNA samples were intact and free of DNA contamination. We added the descriptions of RNA samples integrity detection in the revised manuscript (line 120-122).

L128: The experiment conditions of qRT-PCR assay should be written in the text or Supplementary Tables.

Response: Thanks for your suggestions. According to your suggestion, we have added the experiment conditions of qRT-PCR assay in the Materials and Methods in the revised manuscript (line 138-142).

Fig.2 and L194. Red light appears to have induced a shift from protein to carbohydrate and lipid. What do you think about the former change?

Response: Thanks for your comments. The decrease in protein content under red light may be due to protein degradation, as we found that autophagy-related genes were significantly upregulated and enriched in the GO terms under red light. Protein degradation led to carbon metabolic flux to carbohydrates and lipids, which increased carbohydrates and lipids under red light. We added the corresponding discussion in the revised manuscript (line 250-252).

L218-219: Is it possible that the carbon flux could have been altered by a gene whose expression was reduced?

Response: Thank you for pointing this out. Indeed, it is possible that the carbon flux is altered by a gene whose expression is reduced. However, our study did not identify specific genes. Therefore, in combination with the growth and metabolites results, this study primarily focused on analyzing the biological functions of upregulated genes under red and blue lights.

Fig.3D&E The KEGG enrichment analysis appears to be less consistent with the results in Figure 2. Were protein or lipid synthesis related genes enriched under blue or red light? 

Response: Thanks for your comments. The top most significantly up-regulated KEGG pathways were: photosynthesis-antenna proteins, photosynthesis, and fatty acid biosynthesis in the blue or red light (Fig. 3D). Ribosomes are where proteins are synthesized, and the number of ribosomes determines the amount of protein. Ribosome was also significantly enriched in the up-regulated KEGG pathways in the blue vs white light group. Moreover, most of the genes in these KEGG pathways were up-regulated. These results indicated that blue or red light promoted the growth and lipid synthesis of Chlorella pyrenoidosa. In addition, KEGG and GO enrichment analysis showed that blue light promoted ribosome synthesis, while red light promoted autophagy. Therefore, these analytical data are roughly consistent with the results in Figure 2.

Reviewer 2 Report

Comments and Suggestions for Authors

Authors investigate the hypothesis that the light wavelength influences the biomass composition of the green microalgae in question, by analyzing the biomass composition of algae cultivated under white, red and blue light produced by appropriate LEDs. For this purpose, authors analyzed transcriptome and metabolome differences in a green microalgae strain Chlorella pyrenoidosa when separately cultivated under white, red and blue light. Cultivation under red and blue light promoted growth and biomass accumulation, cultivation under blue light reduced the content of the total carbohydrate while increasing the contents of protein and lipids and cultivation under red light decreased the content of protein and increased the contents of carbohydrate and lipids. Blue and red light influenced also distinctly the fatty acid composition in the lipid fraction in a different way, red light enhancing the synthesis of C16 FA and blue light enhancing the synthesis of C18 FA. Transcriptome analysis of genes expressed under white, red and blue light shows results consistent with the metabolome analysis.

Comments:

- P2/P3, L 91-99:  the light intensity is always set to 200 micromol/m2/s. Authors do not specify with which instrument and in which range the light intensity was measured, was it PAR (400 to 700 nm) or the complete LED spectrum (in the case of the white LED), or simply values taken from the respective LED specification intensity/current/voltage data supplied by the manufacturer? Please supply this information.

A general comment: the spectrum of white light produced by a LED consists in most cases of two peaks, a narrow "blue" one and a wider "green" one going up to red. It means that when measuring the intensity, e.g., in the PAR range the energy supplied by the white LED in the range of the blue or red LED as given in the manuscript is (substantially) lower than 200 micromol/m2/s. It would be interesting to compare the intensities provided by the blue and red LEDs used in experiments (given as 200 micromol/m2/s) to intensities provided by the white LEDs in the same spectral range, blue or red. Could authors please supply this information (e.g., by measuring the white spectrum and integrating the values in the respective red or blue range given by the color LEDs)? Maybe also briefly discussing the possibility that the observed effects are simply the result of higher blue/red light intensities?

- I'd suggest to include the article of Patelou et al., Algal Research 45(2020) 101735, doi: 10.1016/j.algal.2019.101735, in the cited references (in Introduction and Results/Discussion). This article deals with the same topic as the reviewed manuscript so I think it should be definitely mentioned.

- P8/Fig.4: what are QC samples?

Author Response

- P2/P3, L 91-99:  the light intensity is always set to 200 micromol/m2/s. Authors do not specify with which instrument and in which range the light intensity was measured, was it PAR (400 to 700 nm) or the complete LED spectrum (in the case of the white LED), or simply values taken from the respective LED specification intensity/current/voltage data supplied by the manufacturer? Please supply this information.

Response: Thanks for your comments. According to your suggestions, we have added instrument and ranges for light intensity measurement in the Materials and Methods in the revised manuscript (line 99-101).

A general comment: the spectrum of white light produced by a LED consists in most cases of two peaks, a narrow "blue" one and a wider "green" one going up to red. It means that when measuring the intensity, e.g., in the PAR range the energy supplied by the white LED in the range of the blue or red LED as given in the manuscript is (substantially) lower than 200 micromol/m2/s. It would be interesting to compare the intensities provided by the blue and red LEDs used in experiments (given as 200 micromol/m2/s) to intensities provided by the white LEDs in the same spectral range, blue or red. Could authors please supply this information (e.g., by measuring the white spectrum and integrating the values in the respective red or blue range given by the color LEDs)? Maybe also briefly discussing the possibility that the observed effects are simply the result of higher blue/red light intensities?

Response: Thanks very much for your comments. This manuscript mainly compared the effects of white light, red light and blue light on Chlorella pyrenoidosa under the same light intensity. Indeed, there is a problem that white light also contains some blue light and red light, which makes the variables not unique. The best way is to adjust white light to darkness. However, because microalgae need light to grow normally, this manuscript set white light as a control. In fact, many papers use white light as a control to compare the effects of different light qualities on microalgae growth. As you mentioned, it would be interesting to compare the effects of microalgae in blue or red provided by white LEDs to the same intensities provided by red or blue LED lights alone, which will be our next research work.

- I'd suggest to include the article of Patelou et al., Algal Research 45(2020) 101735, doi: 10.1016/j.algal.2019.101735, in the cited references (in Introduction and Results/Discussion). This article deals with the same topic as the reviewed manuscript so I think it should be definitely mentioned.

Response: Thanks for your comments. We have cited the reference in the reviewed manuscript.

- P8/Fig.4: what are QC samples?

Response: Thanks for your comments. Quality control (QC) samples are composed of equal volumes of test samples, containing all information on metabolites of test samples. QC samples are used to assess the accuracy, stability, and repeatability of experiments. We added the descriptions of QC samples preparation in the Materials and Methods in the revised manuscript (line 152-154).